# Changes in functioning and health during the first 6-months of the COVID-19 pandemic among individuals with a spinal cord injury

Ethan Simpson[1,2,3], William C. Miller [1,2,3]*, Julia Schmidt[1,2], Jaimie Borisoff[3,4], W. Ben Mortenson[1,2,3]

1 Department of Occupational Science & Occupational Therapy, Faculty of Medicine, University of British Columbia, Vancouver, British Columbia, Canada, 2 GF Strong Rehabilitation Research Program, Vancouver, British Columbia, Canada, 3 International Collaboration on Repair Discoveries, Vancouver, British Columbia, Canada, 4 Rehabilitation Engineering Design, British Columbia Institute of Technology, Burnaby, British Columbia, Canada

* bill.miller@ubc.ca

**Data Availability Statement:** All relevant data are within the manuscript and its Supporting information files.

## Abstract

### Study design

Single-cohort longitudinal survey design.

### Objectives

To identify what ongoing impact the COVID-19 pandemic has on functioning and health in individuals with SCI. Using the ICF model as a guide, outcome measures were chosen to explore potential constructs and aspects of health and functioning which may have been affected by regulations.

### Setting

Online, Canada.

### Methods

Participants provided demographic and clinical characteristics at baseline. They completed standardized online measures at three time points, each roughly one month apart (June, July, and August of 2020). The measures assessed mental health, resilience, boredom, social support, technology use, life space, and participation. Repeated measures ANOVAs were used to identify longitudinal changes for each measure.

### Results

We collected data from 21 participants with SCI (mean age 54 years, 12 male). We found a large effect size for participation ($\eta^2 = 0.20$), which increased over time. We also found medium effect sizes in both anxiety ($\eta^2 = 0.12$) and social network usage ($\eta^2 = 0.12$). Anxiety decreased over time and social networking usage fluctuated slightly but with an increase from time point one to time point two.

**Funding:** The author(s) received no specific funding for this work.

## Conclusion

The results indicate that individuals with spinal cord injury appear to be staying relatively stable during the pandemic with improvements in a few key aspects, such as potentially increased participation and decreased anxiety. The results also suggest that it is important to continue fostering ways for individuals with spinal cord injury to stay connected, engaged, and informed.

## Introduction

The COVID-19 pandemic has required vast changes in how societies operate. The exact guidelines and restrictions in place during the pandemic have not been fixed, however, as fluctuating rates of COVID-19 infection have resulted in measures constantly evolving. For example, in British Columbia, Canada an initial four phase plan was developed [1]. The three data collection timepoints of this study happened to align with the first 3 phases of this plan. Among other guidelines, phase 1 included mandated physical distancing, closing non-essential personal services, postponing non-urgent surgeries, and imposing quarantine requirements. It was mainly older populations accounting for cases during this stage [2]. Phases 2 and 3 saw the reopening of many businesses with newly implemented safety precautions[3]. It saw a shift to predominantly younger populations accounting for cases [2]. A halt to the phases came with increased mask mandating in all public indoor spaces and transport, and enhanced travel related quarantine requirements. The intention of Phase 4 was to be able to allow the gathering of large groups and international tourism again [1].

Although challenging for most, the lengthy current pandemic may be especially arduous for individuals with a disability, such as those with a spinal cord injury. These individuals require a complex management system to limit the multiple potential physical and psychological secondary conditions associated with spinal cord injury. Common secondary conditions include problematic spasticity, pressure ulcers, autonomic dysreflexia, genitourinary dysfunction, and depression [3]. These secondary conditions can reduce quality of life and social participation; and may have been further exacerbated by restrictions imposed to address COVID-19.

A useful way to describe the impact of the pandemic on individuals with spinal cord injury, is with the International Classification of Functioning, Disability and Health (ICF) model. The ICF model provides a framework for organizing and understanding information related to health outcomes, health determinants, and changes in health status and functioning [4]. It shifts the focus from the health condition to functioning to provide a more neutral and broad comparability between individuals. The ICF model conceptualizes functioning as a dynamic interaction between health conditions (disorders or diseases) and contextual factors (environmental factors and personal factors) [4]. Functioning is a multidimensional concept relating to three identified levels, the level of the body, the whole person, and the whole person in a social context. Dysfunction at one or more of these levels indicates disability: impairment in body functions and structures, activity limitations, and participation restrictions [4].

There are various aspects of health and functioning among individuals with spinal cord injury that may have potentially been affected by regulations. The requirement to physically distance has affected social interactions. This may have changed environmental factors such as social support and technology use. Reductions in social support may include difficulty getting caregivers which could have a compounding effect on an individual's ability to perform day-to-day functions [5]. A combined consequence may be a reduction in activities and

participation. A lack of social interaction could lead to issues in body function and structures such as depression [6]. In addition to depression, the pandemic can cause a range of anxiety inducing factors. Personal factors, such as resilience, may then be crucial for determining how individuals handle the potential barriers and challenges that may arise [7].

There is limited research on how the spinal cord injury community are enduring the pandemic and handling fluctuating regulations and restrictions. The aim of this study was, therefore, to use the ICF model as a partial guide and investigate the changes in functioning and health during the first 6-months of the COVID-19 pandemic among community dwelling individuals with a spinal cord injury. The outcome measures used were chosen to explore the constructs and aspects of health and functioning listed above which may have been affected by regulations.

## Materials and methods

To determine the differences and changes experienced by individuals with a spinal cord injury over time as the COVID-19 pandemic progresses, this study used a single-cohort longitudinal survey design. Data were collected from the same sample across three time points. Ethical approval was obtained from the University of British Columbia and Vancouver Coastal Health (H14-01737). Reporting is informed by the STROBE guidelines.

### Participants

Participants needed to be (1) Canadian, (2) 19 years of age or older, (3) able to speak and understand written and spoken English, (4) have access to technology and internet, and (5) have a spinal cord injury. Individuals were excluded if they have moderate to severe cognitive impairment that inhibited their use of the internet. It was based on a brief screening conversation. Recruitment occurred from 1st April– 31st May 2020 by advertisements, by contacting individuals who have previously been involved in our research studies and consented to being contacted about future research, and via word of mouth. All interested and eligible candidates consented to the study.

### Protocol

Those who expressed interest in participating in the study were provided with further elaboration of the study details and a chance to provide their informed written consent. A Qualtrics link to a survey was then sent for participants to complete. The survey collected quantitative data using the outcome measures outlined below. All but three of the measures have been used in spinal cord injury samples before. Participants were asked to complete the survey once a month for three months (June, July, and August of 2020). Following each survey, participants were sent an email that included an e-transfer with c$30.00.

### Outcome measures

The outcome measures are listed below according to the ICF Framework.

**Body functions and structure.** Hospital Anxiety and Depression Scale (HADS): The HADS is a self-report assessment containing 14 items that screen for anxiety and depression related symptoms. The HADS has been used in a variety of populations, including spinal cord injury [8]. It provides subscale scores for anxiety (0–21) and depression (0–21). A higher score indicates greater anxiety and/or depression. Each subscale has a critical cut-off score of $\geq$8 [9]: "A score of 7 or less for non-cases, scores of 8–10 for doubtful cases, and scores of 11 or more for definite cases" (p. 363).

**Activity/Participation.** Keele Assessment of Participation (KAP): The KAP uses 11-items to assess participation restriction over the previous four weeks. Agreement of the KAP with the 'Impact on Participation and Autonomy' and the 'Reintegration to Normal Living' was used to indicate appropriateness. A mean percentage agreement for corresponding items of 87.7% and 79.3% respectively [10]. There is limited data on its use with spinal cord injury. A total score is calculated (0–11). A higher score indicates more restriction of participation.

Life Space Assessment (LSA): The LSA asks participants to report how frequently and far they have travelled out of the room in which they sleep during the previous four weeks. Appropriateness of the measure is deemed by comparing the LSA to the Reintegration to Normal Living Index with Spearman's rho correlations ranging from 0.509–0.538. The ICC for test-retest reliability nine days apart in participants with spinal cord injury has been reported at 0.876 [11]. A total score is calculated (0–120) by multiplying the life-space level (1–5) by the frequency of mobility (0–4) and level of assistance needed (1–2). A higher score indicates more expansive life space.

Multidimensional State Boredom Scale (MSBS): The MSBS is a 29-item measure of state boredom [12]. It provides a total score (29–203). It also includes subscales targeting disengagement (10–70), high arousal (5–35), inattention (4–28), low arousal (5–35) and time perception (5–35) [12].

**Environmental factors.** Multidimensional Scale of Perceived Social Support (MSPSS): The MSPSS is a 12-item measure of social support. It has a reported Cronbach's coefficient alpha of 0.88 and a 2-month test-retest reliability score of 0.85 with university students [13]. It provides a total score (12–84) with subscales focusing on the source of support; family (4–28), friends (4–28), significant other (4–28). A higher score shows more social support. Total scores of 12–35 are considered low, 36–60 medium, and 61–84 high [13].

Social Networking Usage Questionnaire (SNUQ): The Social Networking Usage Questionnaire is a 19-item scale of social networking usage. Internal consistency is indicated by a Cronbach's coefficient alpha of 0.83 with university students [14]. In our study we had one item missing from the measure. It provides a total score (19–95). A higher score indicates more social networking usage.

**Personal factors.** Demographic Information: We collected age, location of birth, sex, living arrangement, care, education, income, employment status, onset of spinal cord injury, type of injury, diagnosis, and whether they have tested positive for COVID-19.

Connor-Davidson Resilience Scale 25 (CD-RISC-25): The CD-RISC-25 is a 25-item scale assessing resilience and has been deemed appropriate for use with individuals with spinal cord injury [15]. Each item is rated on a 5-point scale and provides a total score (0–100). A higher score represents greater resilience.

Technology Readiness Index (TRI 2.0): The TRI 2.0 is a 16-item measure of an individual's technology readiness. Despite its widespread use, there is limited research regarding the reliability and validity of the TRI 2.0. It provides subscale scores for optimism, innovativeness, discomfort, and insecurity [16]. Average scores are calculated for each (0–5). When calculating an overall TRI 2.0 score, discomfort and insecurity are reverse coded before an average score across the four subscales is calculated.

## Analyses

Descriptive characteristics were used to describe the sample. Outcome measure data were imported to SPSS, and univariate statistics were used to account for the number and percent of missing data in each measure. The patterns of missing values for each measure were evaluated to determine missing data mechanisms. Provided that values were missing at random or

completely at random, and the percentage of the missing values was less than 30%, we imputed the missing values using a multiple imputation technique. The multiple imputation was used to compute five plausible values for each missing value. The missing values were then replaced by the mean of the five plausible values. Multiple imputation analyses were run for each measure that contained missing values in each group separately. Once imputation was completed, individual Repeated Measures ANOVAs were performed for each measure to explore changes in the sample across time. Statistical significance was accepted at the $p \leq 0.05$ level. Effect sizes were calculated for further depth of exploration. Cohen classifies the effect sizes ($\eta^2$) as small (0.01), medium (0.06) or large (0.14) [17].

## Results

The demographic characteristics of the sample are outlined in Table 1. Data were collected from 21 individuals with spinal cord injury. There were initially 22 participants, but one participant did not complete sufficient data collection and, hence, their data were not used for the analyses. The mean age was 54 years, and 9 participants were female.

Table 2 outlines the results of the Repeated Measures ANOVAs for each measure. No changes were statistically significant. However, there was a large effect size for the KAP ($\eta^2$ = 0.20), showing less participation restrictions over time. There was a medium effect size for HADS anxiety ($\eta^2$ = 0.12), indicating reduced anxiety over time. There was also a medium effect size for social network usage ($\eta^2$ = 0.12), showing a slight fluctuation in usage but with a clear increase from time point one. All other measures showed small effect sizes. The HADS depression remained relatively stable with mean scores ranging from 5.33 to 5.57 (P = 0.92. The LSA remained constant with a very minor increase of 2.09 between time points 1 and 3 (P = 0.83). The MSBS reported the highest boredom score at time point 1, followed by a reduction at time point 2 and then a slight increase at time point 3, but all time points had large standard deviations (P = 0.76). Levels of perceived social support remained stable across time points as indicated by the MSPSS (P = 0.96). The CD-RISC-25 showed stable levels of resilience across time points, with a very small increase of 1.00 across mean values (P = 0.94). Technology readiness showed a minimal decrease across time points with the mean score reducing from 3.46 at time point 1 to 3.28 at time point 3 (P = 0.60).

The number of participants who were outside of the cut-off scores for the standardized measures can be seen in Table 3. Cut-off scores were identified for the HADS and for the MSPSS. The number of participants above the cut-off score for anxiety reduced from time 1 to time 2. The number of participants above the cut-off score for depression gradually reduced across time points. The number of participants below the cut-off score for the MSPSS remained constant across time points.

## Discussion

There has been limited research to indicate how fluctuating regulations and restrictions have changed the functioning and health of the spinal cord injury community. There were no statistically significant changes in this study but there were medium/large effect sizes for three measures, each in a different component of the ICF model. The findings suggest that participants were relatively stable across timepoints. As the pandemic was unexpected, however, this study does not have pre-COVID-19 data for the participants. We, therefore, first looked for relatively comparable pre-COVID-19 spinal cord injury studies to see whether the outcome measure scores in this study followed pre-COVID-19 trends. The discussion will then explore the measures which showed a medium to large effect size.

**Table 1. Sample demographic information (N = 21).**

| Demographic factor | Responses | X±SD or n (%) |
|---|---|---|
| Age | | 53.95±11.31 |
| Birthplace | Canada | 14 (67) |
| | Other | 5 (24) |
| | Prefer not to answer | 2 (10) |
| Sex | Female | 9 (43) |
| | Male | 11 (52) |
| | Prefer not to answer | 1 (5) |
| Live in assisted living | No | 21 (100) |
| Live alone | Yes | 7 (33) |
| Provide care | Yes | 2 (10) |
| Receive care | Yes | 7 (33) |
| Education | Graduated from high school/GED | 1 (5) |
| | Some/graduated college/trade school/ university | 17 (81) |
| | Some/graduated post-graduate school | 3 (14) |
| Household income | Less than $14,999 | 2 (10) |
| | $15,000 to $44,999 | 5 (24) |
| | $45,000 to $74,999 | 6 (29) |
| | Greater than $75,000 | 2 (10) |
| | Prefer not to answer | 6 (29) |
| Employment status | Employed full-time | 3 (14) |
| | Employed part-time | 3 (14) |
| | Self employed | 2 (10) |
| | On disability assistance | 6 (29) |
| | Retired | 5 (24) |
| | Unemployed | 1 (5) |
| | Other | 1 (5) |
| Time with disability | Since birth | 2 (10) |
| | Since childhood | 1 (5) |
| | Since adolescence | 4 (19) |
| | Since adulthood | 6 (29) |
| | Later in life | 8 (38) |
| Type of injury | Complete | 6 (29) |
| | Incomplete | 14 (67) |
| | Prefer not to answer | 1 (5) |
| Diagnosis | Paraplegia | 9 (43) |
| | Tetraplegia | 6 (29) |
| | Other (e.g., spina bifida) | 6 (29) |
| Ambulatory status | Ambulatory | 3 (14) |
| | Non-ambulatory | 15 (71) |
| | Other | 3 (14) |
| Tested COVID-19 positive | Yes | 0 (0) |

## Measures with a small effect size

A Dutch study conducted in 2019 recorded CD-RISC-25 scores of 69 compared to the scores of 71–72 in this study, suggesting similar resilience levels [15]. For the MSPSS, values of 61–62 in this study compared with 61.4 from a study in Iran, indicate stable levels of perceived social

**Table 2. Standardized measures listed alphabetically and repeated measures ANOVAs.**

| Measure | Time 1 | Time 2 | Time 3 | P | F | $\eta^2$ |
|---|---|---|---|---|---|---|
| | Mean ± Standard Deviation | | | | | |
| Resilience (CD-RISC-25) | 71.19 ± 15.36 | 71.52 ± 16.11 | 72.19 ± 15.87 | 0.94 | 0.07 | 0.01 |
| Anxiety (HADS) | 7.48 ± 5.16 | 6.29 ± 4.71 | 5.95 ± 4.71 | 0.30 | 1.28 | 0.12 |
| Depression (HADS) | 5.43 ± 4.47 | 5.33 ± 4.51 | 5.57 ± 4.87 | 0.92 | 0.09 | 0.01 |
| Participation (KAP) | 3.10 ± 2.51 | 2.81 ± 2.29 | 2.10 ± 2.61 | 0.12 | 2.34 | 0.20 |
| Life Space (LSA) | 42.88 ± 13.83 | 42.69 ± 15.17 | 44.97 ± 18.09 | 0.83 | 0.19 | 0.02 |
| Support (MSPSS) | 61.19 ± 15.75 | 62.33 ± 18.21 | 61.24 ± 18.09 | 0.96 | 0.04 | 0.01 |
| Boredom (MSBS) | 98.71 ± 44.24 | 93.48 ± 40.07 | 95.48 ± 42.00 | 0.76 | 0.28 | 0.03 |
| Social Networking (SNUQ) | 62.29 ± 16.50 | 67.33 ± 8.71 | 66.10 ± 13.00 | 0.29 | 1.31 | 0.12 |
| Technology Readiness (TRI 2.0) | 3.46 ± 0.57 | 3.34 ± 0.50 | 3.28 ± 0.73 | 0.60 | 0.52 | 0.05 |

**Table 3. Number of participants outside cut-off score.**

| Measure | Cut-off score | n (%) of participants outside cut-off score | | |
|---|---|---|---|---|
| | | Time 1 | Time 2 | Time 3 |
| Anxiety (HADS) | ≥8 | 10 (48) | 6 (29) | 6 (29) |
| Depression (HADS) | ≥8 | 7 (33) | 6 (29) | 4 (19) |
| Support (MSPSS) | ≤35 | 2 (10) | 2 (10) | 2 (10) |

support [18]. Only two participants in this study scored below the cut-off score for the MSPSS indicating that 90% of participants had medium or high levels of social support [13]. For the TRI 2.0, this study had scores of 3.28–3.46 compared to 3.5 in a Canadian study, indicating consistent levels of technology readiness [19]. For HADS depression, the scores of 5.33–5.57 in this study are comparable with 5.5 from a study in the UK [8]. Based on the cut-off scores, however, this study did have depression rates that exceeded the standard 1 in 5 for individuals with spinal cord injury at time points 1 and 2 [20]. Rates of 33% and 29% respectively of participants exceeding the cut-off scores indicate that the COVID-19 pandemic may have had an impact on inducing depression [9]. For the LSA, compared to a score of 66 in a study in the United States, this study only reported scores of 42.69–44.97 [11]. This suggests that participants might not be traveling as far from home compared to pre-COVID-19 and this could be indicative of guidelines and restrictions implemented during the pandemic reducing life space. No previous spinal cord injury data could be found for the MSBS, KAP, or SNUQ.

## Measures with medium to large effect size

In the body function and structure component of the ICF model, there seemed to be a decrease in anxiety across time points. Potential sources of COVID-19 related anxiety, for individuals with spinal cord injury, were suggested to include contracting the virus, finding sufficient caretakers, accessing and maintaining specialized medical care and equipment, getting to appointments, and a potential inability to self-quarantine [5]. Anxiety itself can be defined as "a tense unsettling anticipation of a threatening but formless event, a feeling of uneasy suspense" [21]. This relates to uncertainty as outlined by components of the Entropy Model of Uncertainty, explaining that uncertainty can be subjectively experienced as anxiety and that individuals are motivated to keep uncertainty at a manageable level [22]. The changes to daily routines

associated with the COVID-19 pandemic may have elevated the level of uncertainty for some individuals, resulting in a higher initial anxiety value. As new information was discovered and released, aspects of uncertainty may have been addressed, showing a gradual decrease in anxiety. From close to the critical cut-off point of 8 at time 1 with 7.48, time 2 and 3 were back within values seen for individuals with spinal cord injury by previous research pre-COVID-19, with 6.29–5.95 compared to 6.9 in a study in the UK [8]. Over time, participants may have also learned to adopt coping behaviors to reduce COVID-19 related anxiety inducing factors. Research in the Spanish general population has indicated that these coping behaviors include following a healthy diet, not reading about COVID-19 news too often, pursuing hobbies, and staying outdoors [23]. Of the participants, 29% did still exceed the cut-off score for anxiety at time points 2 and 3, however, indicating that COVID-19 related anxiety may have continued for some [9].

For the activities and participation component of the ICF model, there was an increase in participation during the initial period of the pandemic. One explanation for this could simply be changes to restrictions over time. The sample of this study was British Columbia based where the government implemented the four-phase plan mentioned previously. Following the closure of all non-essential businesses during Phase 1, the 2nd and 3rd phases saw the reopening of many businesses but with new safety precautions in place [1]. As the data collection points happened to align relatively well with the first three phases, this could indicate that as more facilities reopened, there was more room for participation among the sample. Another reason for seeing an increase in participation is that the participants may have gradually adapted to finding new activities that they could still engage with under guidelines and regulations. Using the broader population as an example, there has been an increase during the pandemic in home activities, such as baking and gardening. While many shifted to a more sedentary lifestyle and weight gain, others shifted to finding physically distanced exercises, such as home-workouts, running, biking, and hiking [24]. Although individuals with spinal cord injury would have to do adapted versions of these activities in many cases, it is still conceivable that they have followed the same trend, shifting from their old activities to new ones.

Within the environmental factors component of the ICF model, the results suggested a general increase in social networking usage, which may have helped combat some of the impacts of restrictions. Social networking refers to "the use of websites and other internet services to communicate with other people and make friends" [25]. A small study into the information technology usage of individuals with spinal cord injury pre-COVID-19 found that 80% of their participants used social networking websites as they provide a convenient means to maintain and make connections. Based on open-ended survey responses, social support is found to facilitate resilience post-spinal cord injury [26]. With other avenues of social support threatened by physical distancing and other restrictions, social networking may provide a means of staying connected. Not only has an increased use of social networking aided general communication, but it is also being increasingly used for work, education, research, and information dissemination [27].

The longitudinal nature of this study allowed us to follow participants across time points and analyse any differences in their behavior or potential level of coping as the pandemic progressed. Future studies could continue to monitor the well-being of individuals with spinal cord injury at various time points. Future research designs could analyse direct comparisons between sub-groups within spinal cord injury or between spinal cord injury and other populations during the pandemic and ultimately post-pandemic.

## Limitations

Limitations of the study could relate to study design, sample size, and statistical analysis. The longitudinal nature of the data prevents causal inferences from being made. The limited

sample size likely increases the risk of type II error when identifying statistical significance, but the execution of multiple comparisons likely increases the risk of type I error. Therefore, analysis of effect sizes was particularly important. There are some potential issues with generalizability of the study findings as there was a higher ratio of female to male participants compared to broader Canadian spinal cord injury demographics. Furthermore, all participants resided in British Columbia so some COVID-19 responses may differ elsewhere in the world, compared to BC's four phase plan, and have different impacts on individuals with spinal cord injuries. Individuals were also excluded if they had cognitive impairments or did not have access to technology, so they may have experienced the initial stages of the pandemic differently. Although many relevant variables were measured, there may have been unmeasured confounding variables. For example, some changes may have been caused by seasonal climatic variations rather than pandemic related restrictions.

## Conclusion

In conclusion, the results show that individuals with spinal cord injury appear to be staying relatively stable during the pandemic with gradual improvements in a few key aspects. Their potentially increased participation and decreased anxiety are a positive sign for their psychological and physiological health. The results suggest that it is important to continue fostering ways for individuals with spinal cord injury to stay connected, engaged, and informed. It is also still important to mitigate the multiple challenges they already face that could be compounded by the pandemic.

## Supporting information

**S1 Data.**
(XLSX)

## Author Contributions

**Conceptualization:** Ethan Simpson, William C. Miller, Julia Schmidt, Jaimie Borisoff, W. Ben Mortenson.

**Data curation:** Ethan Simpson, W. Ben Mortenson.

**Formal analysis:** Ethan Simpson, W. Ben Mortenson.

**Investigation:** Ethan Simpson.

**Methodology:** Ethan Simpson, William C. Miller, Julia Schmidt, Jaimie Borisoff, W. Ben Mortenson.

**Project administration:** Ethan Simpson, W. Ben Mortenson.

**Supervision:** W. Ben Mortenson.

**Writing – original draft:** Ethan Simpson, W. Ben Mortenson.

**Writing – review & editing:** Ethan Simpson, William C. Miller, Julia Schmidt, Jaimie Borisoff, W. Ben Mortenson.

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
