## [Decision Letter · Decision Letter 0]

13 Dec 2023

PONE-D-23-30086Changes in functioning and health during the first 6-months of the COVID-19 pandemic among individuals with a spinal cord injuryPLOS ONE

Dear Dr. Miller,

Thank you for submitting your manuscript to PLOS ONE. After careful consideration, we feel that it has merit but does not fully meet PLOS ONE’s publication criteria as it currently stands. Therefore, we invite you to submit a revised version of the manuscript that addresses the points raised during the review process.

Dear authors,

Thank you for submitting your manuscript to PLOS ONE. After review, it is clear that your study on spinal cord injury during the COVID-19 pandemic is of potential interest, but requires major revisions. You can find the detailed reviews below. Key areas to address include:

Enhancing clarity and detail in the abstract and introduction, particularly regarding the study's timeframe and scope.Providing more comprehensive information on your methodology, especially in terms of participant recruitment, data handling, and the definition and implications of incomplete SCI.Strengthening the discussion section by directly linking it to your results and offering a clearer, more focused analysis without overstating findings.==============================

We look forward to receiving your revised manuscript.

Kind regards,

Nicola Diviani

Academic Editor

PLOS ONE

4. Please ensure that you include a title page within your main document. You should list all authors and all affiliations as per our author instructions and clearly indicate the corresponding author.

5. Please amend your authorship list in your manuscript file to include author WC Miller, Ethan Simpson, Julia Schmidt, Jaimie Borisoff and W. Ben Mortenson.

7. Please include your tables as part of your main manuscript and remove the individual files. Please note that supplementary tables (should remain/ be uploaded) as separate ""supporting information"" files".

Reviewers' comments:

Reviewer's Responses to Questions

**Comments to the Author**

1. Is the manuscript technically sound, and do the data support the conclusions?

Reviewer #1: No

Reviewer #2: Yes

2. Has the statistical analysis been performed appropriately and rigorously? 

Reviewer #1: I Don't Know

Reviewer #2: I Don't Know

3. Have the authors made all data underlying the findings in their manuscript fully available?

Reviewer #1: No

Reviewer #2: No

4. Is the manuscript presented in an intelligible fashion and written in standard English?

Reviewer #1: Yes

Reviewer #2: Yes

5. Review Comments to the Author

Reviewer #1: The authors sought to examine changes in function and health among 21 adults with spinal cord injury (SCI) from British Columbia in Canada during the early months of the COVID-19 pandemic (i.e., June, July, August 2020). The paper would be improved by providing more rigorous detail to sample recruitment, missing data specific to the 3 surveys and specific items, and not over-stating findings since no results were significant and the timeframe was during 3 consecutive, early months of the pandemic.

- The abstract should include details about the months of the survey

- The first half of the first paragraph of the paper is too in the weeds of the COVID pandemic (e.g., the first two sentences are not needed) - which distracts from the purpose of the paper. It would be sufficient to discuss how the COVID-19 pandemic impacted the world quickly and dramatically (with relevant cites). Citing specific case numbers becomes dated with a point in time and it's not necessarily to discuss specific variants (especially since those variants were out after the timeframe of the study).

- The authors should make it clear that the 4 phases of the early pandemic that they are discussing map to the timeframe of their data collection - if it does.

- What does an incomplete SCI mean and how was this assessed? What does this mean for the findings - that 14 of the 21 participants had an incomplete SCI?

- More information is needed about recruitment - how many initially agreed to participate but did not consent to the study?

- The authors mentioned that they did imputation for missing data but do not provide specifics. Did all 21 participants complete all 3 surveys? More specific detail is needed about missing data and decisions about imputation.

- The tables were missing from the paper

- The discussion spends a lot of time listing studies from other countries - more discussion is needed about if these are appropriate comparisons.

Reviewer #2: The stated reason for this research was not initially clear to me. Somewhat generic. There are two points emphasised in the abstract conclusion. That environmental changes from regulations and restrictions affected individuals with SCI. Therefore "need for continued flow of information and for adaptive activities" Lacks clarity in itself.

The discussion section should circle back to the results in a more descriptive fashion. Having the tables and data available might have made it easier. Not sure. That the individuals with SCI fared relatively well except perhaps a small number with depression and or anxiety, is interesting in and of itself and maybe more discussion of this may be in order? So a few tweaks to emphasize the results in a clear descriptive statement would make reading it more readily understandable. This would draw the reader in and pique the desire to assess the validity of the conclusions.

6. PLOS authors have the option to publish the peer review history of their article (what does this mean?). If published, this will include your full peer review and any attached files.

Reviewer #1: No

Reviewer #2: No

---

## [Author Response · Author response to Decision Letter 0]

29 Jan 2024

Thank you for your feedback and suggestions. Please find our comments for each point below.

Editor

Enhancing clarity and detail in the abstract and introduction, particularly regarding the study's timeframe and scope.

We have included clarifying statements regarding both timeframe and objectives in the abstract and introduction.

Providing more comprehensive information on your methodology, especially in terms of participant recruitment, data handling, and the definition and implications of incomplete SCI.

We have addressed these items in the methods section and responded to the reviewer below.

Strengthening the discussion section by directly linking it to your results and offering a clearer, more focused analysis without overstating findings.

We have tweaked the discussion to highlight areas of interest and potential reasons but have avoided making absolute statements.

Reviewer #1: The authors sought to examine changes in function and health among 21 adults with spinal cord injury (SCI) from British Columbia in Canada during the early months of the COVID-19 pandemic (i.e., June, July, August 2020). The paper would be improved by providing more rigorous detail to sample recruitment, missing data specific to the 3 surveys and specific items, and not over-stating findings since no results were significant and the timeframe was during 3 consecutive, early months of the pandemic.

- The abstract should include details about the months of the survey

We have included the months when surveys were completed on line 39.

- The first half of the first paragraph of the paper is too in the weeds of the COVID pandemic (e.g., the first two sentences are not needed) - which distracts from the purpose of the paper. It would be sufficient to discuss how the COVID-19 pandemic impacted the world quickly and dramatically (with relevant cites). Citing specific case numbers becomes dated with a point in time and it's not necessarily to discuss specific variants (especially since those variants were out after the timeframe of the study).

We have removed the first three sentences and amended the fourth so that the paper starts with more relevant information (Line 56).

- The authors should make it clear that the 4 phases of the early pandemic that they are discussing map to the timeframe of their data collection - if it does.

We have included that the three data collection timepoints of this study happened to align with the first 3 phases of BC’s plan (Line 59).

- What does an incomplete SCI mean and how was this assessed? What does this mean for the findings - that 14 of the 21 participants had an incomplete SCI?

An incomplete SCI means that the brain’s ability to transmit signals below the site of injury is not fully removed. Individuals with an incomplete SCI may still have some level of feeling and/or function below the injury site. It is the more prevalent type of SCI, which is consistent with our sample. Participants were simply asked whether they had a complete or incomplete SCI during the questionnaire. We have retained the information in the demographic table (Table 1) but removed it from the text. 

- More information is needed about recruitment - how many initially agreed to participate but did not consent to the study?

We have increased the information around recruitment (Line 123). 

- The authors mentioned that they did imputation for missing data but do not provide specifics. Did all 21 participants complete all 3 surveys? More specific detail is needed about missing data and decisions about imputation.

We have clarified the imputation approach starting on line 203: “Data were imported to SPSS, and univariate statistics were used to account for the number and percent of missing data in each measure. The patterns of missing values for each measure were evaluated to determine missing data mechanisms. Provided that values were missing at random or completely at random, and the percentage of the missing values was less than 30%, we imputed the missing values using a multiple imputation technique. The multiple imputation was used to compute five plausible values for each missing value. The missing values were then replaced by the mean of the five plausible values. Multiple imputation analyses were run for each measure that contained missing values in each group separately.”

We have also included a statement on line 219 clarifying that: “There were initially 22 participants, but one participant did not complete sufficient data collection and, hence, their data were not used for the analyses.”

- The tables were missing from the paper

The three tables have now been added to the results section of the main body of the manuscript.

- The discussion spends a lot of time listing studies from other countries - more discussion is needed about if these are appropriate comparisons.

We have provided more of a structure to the discussion, and partially shortened this section so as not to distract from the other discussion elements.

Reviewer #2: The stated reason for this research was not initially clear to me. Somewhat generic. 

The objective is hopefully clearer now (Line 103).

There are two points emphasised in the abstract conclusion. That environmental changes from regulations and restrictions affected individuals with SCI. Therefore "need for continued flow of information and for adaptive activities" Lacks clarity in itself.

We have amended the abstract conclusion to be more in line with the findings (Line 47).

The discussion section should circle back to the results in a more descriptive fashion. Having the tables and data available might have made it easier. Not sure. That the individuals with SCI fared relatively well except perhaps a small number with depression and or anxiety, is interesting in and of itself and maybe more discussion of this may be in order? So a few tweaks to emphasize the results in a clear descriptive statement would make reading it more readily understandable. This would draw the reader in and pique the desire to assess the validity of the conclusions.

We have tweaked the discussion to highlight areas of interest and potential reasons but have avoided making absolute statements.

---

## [Decision Letter · Decision Letter 1]

13 Feb 2024

Changes in functioning and health during the first 6-months of the COVID-19 pandemic among individuals with a spinal cord injury

PONE-D-23-30086R1

Dear Dr. Miller,

We’re pleased to inform you that your manuscript has been judged scientifically suitable for publication and will be formally accepted for publication once it meets all outstanding technical requirements.

Kind regards,

Nicola Diviani

Academic Editor

PLOS ONE

Additional Editor Comments (optional):

Reviewers' comments:

Reviewer's Responses to Questions

**Comments to the Author**

1. If the authors have adequately addressed your comments raised in a previous round of review and you feel that this manuscript is now acceptable for publication, you may indicate that here to bypass the “Comments to the Author” section, enter your conflict of interest statement in the “Confidential to Editor” section, and submit your "Accept" recommendation.

Reviewer #1: All comments have been addressed

2. Is the manuscript technically sound, and do the data support the conclusions?

Reviewer #1: Yes

3. Has the statistical analysis been performed appropriately and rigorously? 

Reviewer #1: Yes

4. Have the authors made all data underlying the findings in their manuscript fully available?

Reviewer #1: Yes

5. Is the manuscript presented in an intelligible fashion and written in standard English?

Reviewer #1: Yes

6. Review Comments to the Author

Reviewer #1: The authors have done a good job addressing all previous comments from the reviewers and the editor.

7. PLOS authors have the option to publish the peer review history of their article (what does this mean?). If published, this will include your full peer review and any attached files.

Reviewer #1: No

---

## [Editor Report · Acceptance letter]

28 Feb 2024

PONE-D-23-30086R1 

PLOS ONE

Dear Dr. Miller, 

I'm pleased to inform you that your manuscript has been deemed suitable for publication in PLOS ONE. Congratulations! Your manuscript is now being handed over to our production team.

Kind regards, 

on behalf of

Dr. Nicola Diviani 

Academic Editor

PLOS ONE